# External Validation of a Predictive Model for Thyroid Cancer Risk with Decision Curve Analysis

**DOI:** 10.3390/diagnostics15060686

**Published:** 2025-03-11

**Authors:** Juan Jesús Fernández Alba, Florentino Carral, Carmen Ayala Ortega, Jose Diego Santotoribio, María Castillo Lara, Carmen González Macías

**Affiliations:** 1Department of Obstetrics and Gynaecology, University Hospital of Puerto Real, 11-510 Cadiz, Spain; mariacastillo.747@gmail.com (M.C.L.); carmengonzalezmacias1973@gmail.com (C.G.M.); 2Institute of Research and Innovation in Biomedical Sciences of the Province of Cadiz, University Hospital ‘Puerta del Mar’, University of Cadiz, 11-009 Cadiz, Spain; josediego.santotoribio@uca.es; 3Department of Endocrinology and Nutrition, University Hospital of Puerto Real, 11-510 Cadiz, Spain; florencarral@hotmail.com (F.C.); carmenayalaor@hotmail.com (C.A.O.); 4Laboratory Medicine Department, University Hospital of Puerto Real, 11-510 Cadiz, Spain

**Keywords:** thyroid cancer, predictive model, decision curve analysis, papillary thyroid carcinoma (PTC), thyroid nodules, external validation, artificial intelligence (AI), machine learning (ML), diagnostic tools

## Abstract

**Background/Objectives**: Thyroid cancer ranks among the most prevalent endocrine neoplasms, with a significant rise in incidence observed in recent decades, particularly in papillary thyroid carcinoma (PTC). This increase is largely attributed to the enhanced detection of subclinical cancers through advanced imaging techniques and fine-needle aspiration biopsies. The present study aims to externally validate a predictive model previously developed by our group, designed to assess the risk of a thyroid nodule being malignant. **Methods**: By utilizing clinical, analytical, ultrasound, and histological data from patients treated at the Puerto Real University Hospital, this study seeks to evaluate the performance of the predictive model in a distinct dataset and perform a decision curve analysis to ascertain its clinical utility. **Results**: A total of 455 patients with thyroid nodular pathology were studied. Benign nodular pathology was diagnosed in 357 patients (78.46%), while 98 patients (21.54%) presented with a malignant tumor. The most frequent histological type of malignant tumor was papillary cancer (71.4%), followed by follicular cancer (6.1%). Malignant nodules were predominantly solid (95.9%), hypoechogenic (72.4%), with irregular or microlobed borders (36.7%), and associated with suspicious lymph nodes (24.5%). The decision curve analysis confirmed the model’s accuracy and its potential impact on clinical decision-making. **Conclusions**: The external validation of our predictive model demonstrates its robustness and generalizability across different populations and clinical settings. The integration of advanced diagnostic tools, such as AI and ML models, improves the accuracy in distinguishing between benign and malignant nodules, thereby optimizing treatment strategies and minimizing invasive procedures. This approach not only facilitates the early detection of cancer but also helps to avoid unnecessary surgeries and biopsies, ultimately reducing patient morbidity and healthcare costs.

## 1. Introduction

Thyroid cancer ranks among the most prevalent endocrine neoplasms. In the United States, its incidence has surged significantly over recent decades, reaching its zenith around 2014. This rise is largely attributed to the increased detection of subclinical cancers, particularly papillary thyroid carcinoma (PTC), facilitated by the widespread adoption of imaging techniques and fine-needle aspiration biopsies [1,2,3]. In Europe, a similar trend has been observed, with a significant rise in the incidence of thyroid cancer, particularly papillary thyroid carcinoma (PTC), over the past few decades [4,5,6,7].

The early and precise identification of malignant thyroid nodules significantly enhances clinical outcomes and reduces the morbidity associated with unnecessary treatments. The integration of advanced diagnostic tools, such as artificial intelligence (AI) and machine learning (ML) models, improves the accuracy in distinguishing between benign and malignant nodules, thereby optimizing treatment strategies and minimizing invasive procedures. This approach not only facilitates the early detection of cancer but also helps to avoid unnecessary surgeries and biopsies, ultimately reducing patient morbidity and healthcare costs [8,9,10,11,12].

The present study centers on the external validation of a predictive model previously developed by our group [13], which is designed to assess the risk of a thyroid nodule being malignant and is available online at the link https://obgynreference.shinyapps.io/calccdt/ (accessed on 7 February 2025). External validation is a crucial step to confirm the generalizability and robustness of the model across different populations and clinical settings. By utilizing clinical, analytical, ultrasound, and histological data from patients treated at the Puerto Real University Hospital, this study seeks to evaluate the performance of the predictive model in a dataset distinct from that used for its initial development.

Furthermore, a decision curve analysis has been performed to ascertain the clinical utility of the model in routine practice. This method will not only confirm the model’s accuracy but also assess its potential impact on clinical decision-making and the management of patients with thyroid nodular pathology.

## 2. Materials and Methods

In this retrospective study, the clinical, analytical, ultrasound, and histological data of 455 patients treated for thyroidectomy at the Puerto Real University Hospital (Cádiz, Spain) between 2019 and 2023 were analyzed to perform the external validation of a predictive model of the risk of thyroid cancer previously developed by our group [12]. The aim of the study was to evaluate the performance of our predictive model in a dataset distinct from the one used to develop it and to perform a decision curve analysis of the model.

### 2.1. Patients

In our center, all patients with suspected thyroid nodular pathology are evaluated in our endocrinology department through a neck ultrasound conducted in a single session. Patients were selected for thyroid FNA based on the recommendations of the American Thyroid Association (ATA) [14,15], and this procedure was carried out during a subsequent appointment. This local approach has demonstrated cost-efficiency in managing patients with thyroid pathology [16] and has shortened the clinical study period before thyroid surgery. Additionally, it has shown a high diagnostic capability in identifying patients with malignant thyroid nodules before performing thyroid FNA [17].

All patients meeting the ATA criteria underwent FNA, performed by a single endocrinologist. The procedure utilized a 20 mL syringe with a 23G needle, guided by images from a Sonosite Micromax ultrasound scanner (models from 2013 to 2016) and a Hitachi Aloka F37 (model from 2017 to 2018) with a 10–14 MHz transducer. Prior to the puncture, all cases were recorded in a standardized registry system database, which included the variables listed in Table 1. Post-puncture, the results of the thyroid cytology (description and Bethesda category) were added to the registry but were not evaluated in our study. In all cases, the indication for thyroidectomy was established in a joint clinical session with the Department of General Surgery. General criteria included single or multi-nodular goiters with nodules 4 cm or larger, compressive symptoms, thyroid hyperfunction, and nodules with Bethesda V and VI cytology. For nodules with Bethesda III or IV cytology, the indication for thyroidectomy was individualized for each case. The thyroidectomy samples were analyzed by the Pathological Anatomy Department of our center, and thyroid incidentalomas (asymptomatic thyroid tumors smaller than 1 cm discovered incidentally during pathological study) were not considered cases of TC.

### 2.2. Statistical Analysis

To perform the external validation of the predictive model, we followed the steps recommended by Riley et al. [18]. All the statistical procedures were performed using the software R, version 4.3.3 [19].

To make predictions for each patient, we used the backward step logistic regression model previously developed by us [12]. The intercept, variables, and their corresponding coefficients are shown in Table 1. Following the recommendations of Riley et al., all predictions were generated and stored in the test dataset by code. The observed distribution of predictions was summarized and presented as a histogram. Additionally, we calculated the median, interquartile range (IQR), mean, and standard deviation (SD) of the predicted probabilities.

Next, we evaluated the model’s predictive performance. First, we quantified the overall fit by calculating R2, Cox-Snell R2 and the Brier score. To calculate Cox-Snell R2, we used the function nagelkerke() from the package rcompanion for R [20]. The Brier score was obtained by using the calibrationCurves package for R [21,22,23]. To evaluate the agreement between observed and predicted values, we generate a calibration plot using the function val.prob.ci.2 from the package calibrationCurves for R. The calibration plot was complemented by determining the calibration slope, intercept (calibration in the large), and observed/expected ratio. Discrimination capacities of the predictive model were evaluated by calculating the concordance (c) statistic index, where a value of 1 indicates the model has perfect discrimination, while a value of 0.5 indicates the model discriminates no better than chance. Given that our output variable is binary (cancer vs. no cancer), the concordance index is equivalent to the area under the ROC curve. Slope calibration, intercept, and concordance index were calculated using the calibrationCurves package for R. Decision curve analysis was performed by using the function dca() from the dcurves package for R [24].

The study received approval from the Biomedical Research Ethics Committee of Cádiz (Spain) in April 2018. Due to the retrospective nature of the study, informed consent was not required for accessing research data. However, all patients who underwent thyroid FNA and subsequent surgery provided signed informed consent forms for these procedures.

## 3. Results

A total of 455 patients with thyroid nodular pathology were studied. Benign nodular pathology was diagnosed in 357 patients (78.46%), while 98 patients (21.54%) presented with a malignant tumor. The patients with malignant nodules are slightly younger than those with benign nodules (49 ± 19.2 vs. 53 ± 18 years; *p* < 0.05).

The most frequent histological type of malignant tumor was papillary cancer (*n* = 70; 71.4%), followed by follicular cancer (*n* = 6; 6.1%). Other types of cancer were diagnosed in 5 patients. Their main clinical, analytical, and sonographic characteristics are presented in Table 2. The thyroid cancer was more frequent in females than in males (67.3% vs. 32.7%). However, the proportion of males was higher in the group of malignant tumors (32.7% of patients with cancer were males while only 16% of patients with benign nodules were males [*p* < 0.001]).

Additionally, thyroid cancer patients have higher levels of plasma TSH (1.6 ± 1.59 vs. 0.9 ± 1.4 mcU/mL; *p* < 0.001) and have more analytical criteria for autoimmune thyroiditis (positivity of anti-TPOAb and/or anti-TgAb) (33.7% vs. 16.8%; *p* < 0.001) than the patients with benign nodular pathologies.

Regarding the US characteristics, malignant nodules tended to be smaller (21 ± 19.5 vs. 35 ± 17 mm; *p* < 0.001), predominantly solids (95.9% vs. 71.7%; *p* > 0.01), hypoechogenic (72.4% vs. 24.4%; *p* < 0.001), with irregular or microlobed borders (36.7% vs. 3.4%; *p* < 0.001), taller than wide (12.2 vs. 5%; *p* < 0.001), having microcalcifications (37.8% vs. 3.9%; *p* < 0.001), and having associated suspicious lymph nodes (24.5% vs. 2.0% *p* < 0.001).

Figure 1 shows the distribution of the probabilities of malignancy predicted by the model. The median risk predicted was 3.33% (IQR 13.62%), and the mean risk was 14.73% (standard deviation 24.60%).

To assess the predictive performance of the model, we evaluated the overall fit, calibration, discrimination performance, and clinical utility.

The model shows a good overall fit with R2 = 0.4238, Cox and Snell R2 = 0.3988, and Brier score = 0.1170.

Figure 2 shows the calibration plot of the model.

Additionally, the model presents a good calibration performance with a calibration slope of 0.70 (95%CI 0.55–0.85) and an observed/expected ratio of 1.46.

As for the discrimination performance, the concordance (c) statistic was 0.84 (95% CI 0.799–0.888), demonstrating good discrimination capacity. Additionally, an ROC curve was developed (Figure 3). Like the c-statistic, the area under the ROC was 0.84 (95% CI 0.799–0.888). The cutoff point with the smallest distance between the ROC plot and the point (0,1) was 0.0955. This means that we would consider a high risk of malignancy of the nodule when the model predicts a probability of 9.55% or higher. With this cutoff point, we obtain a sensitivity of 71.43% and a specificity of 82.35%. Selecting other cutoff points could maximize sensitivity. For example, with a threshold of 4.94%, the sensitivity rises to 80.61%, but specificity decreases to 70.86%.

To evaluate the clinical utility of the predictive model, we conducted a decision curve analysis. We created two decision curve graphs to assess the net benefit of using the model across different threshold values. Figure 4 presents the decision curve of the model for predicted risks between 0 and 30%. This graph shows that in the lower range of predicted risks, the model does not offer benefits compared to the strategy of treating all patients. In Figure 5, we zoomed in on the graph, focusing on risks between 0 and 10%. This graph shows that, from a threshold of 9%, the model outperforms the strategy of treating all patients.

## 4. Discussion

The external validation performed in this study confirms that the predictive model for thyroid nodule malignancy, previously developed by our group, demonstrates satisfactory predictive capacity (R2 = 0.4238, Cox and Snell R2 = 0.3988, Brier score = 0.1170, and c-score = 0.84). By setting the cut-off point at 0.0955 (when the predicted probability by the model was 9.55% or higher), the sensitivity was 71.43% and the specificity was 82.35%. Furthermore, the decision curve analysis indicates that, from the established cut-off point, the net benefit of employing the predictive model exceeds the strategies of treating all patients or none.

The early detection of thyroid cancer (TC) is crucial for improving patient outcomes, particularly when compared to the limited prognosis associated with advanced thyroid tumors. Early diagnosis allows for timely intervention, which significantly enhances survival rates, especially in cases like medullary thyroid carcinoma (MTC), where early-stage detection can lead to a 90–100% ten-year survival rate. In contrast, advanced stages of TC are linked to a stark decline in prognosis, with survival rates dropping to as low as 17% [25,26].

Aside from numerous studies focused on assessing the risk of malignancy in lymph nodes in thyroid cancer [27], predictive models have shown promise in distinguishing between malignant and benign thyroid nodules, leveraging various data sources such as ultrasound images, clinical data, and laboratory parameters. These models aim to improve diagnostic accuracy and reduce unnecessary procedures [13,28,29,30,31,32,33].

Different approaches and models have been used in recent studies. Focusing on machine learning, the Random Forest algorithm has demonstrated exceptional effectiveness in diagnosing malignant thyroid nodules, surpassing the performance of radiologists in assessments based on conventional ultrasound and real-time elastography [34]. Conversely, the Bagged CART model exhibited remarkable accuracy, achieving a 99.1% success rate in predicting thyroid cancer by utilizing clinical data and ultrasound characteristics [35]. Furthermore, the XGBoost model has demonstrated effectiveness in predicting the malignancy and metastasis of thyroid cancer, achieving an AUC of 0.84 for nodule diagnosis and up to 0.97 for metastasis prediction [36]. In addition, deep learning models such as ThyNet have significantly enhanced the diagnostic performance of radiologists, thereby reducing the necessity for unnecessary fine-needle aspirations [37].

Cao et al. [38] compared a logistic model incorporating clinical, ultrasound (US), and genetic variables with other models based on machine learning. They found that the logistic model exhibited higher AUC values. Specifically, the logistic regression model, utilizing backward stepwise regression, achieved area under the curve (AUC) values of 0.83 in the training cohort and 0.80 in the validation cohort. These values indicate superior discrimination between malignant and benign nodules compared to the machine learning models, which demonstrated moderate performance with AUC values around 0.74 for both the Random Forest and XGBoost models. On the other hand, Zhang et al. [39] developed another logistic model based on demographic, serological, and ultrasound data. These authors reported a ROC AUC of 0.924. However, external validation is required to confirm its effectiveness. In this context, our model demonstrates its advantages by exhibiting a robust performance with a ROC AUC of 0.84 when applied to a population distinct from the one used for its development.

To the best of our knowledge, this is the first study to conduct a decision curve analysis of a predictive model for the malignancy of thyroid nodules. To comprehend the decision curve analysis, it is essential to grasp some key concepts [40]. The *y*-axis represents the benefit, while the *x*-axis denotes the preference. But what does this signify in the specific case we are examining? Let’s consider the scenario. We are dealing with a patient who has a thyroid nodule that may or may not be indicative of thyroid cancer. The decision we are attempting to make is whether to remove the nodule.

To correctly interpret a decision curve analysis, it is essential to recognize that the decision can vary based on the patient’s preferences as well as those of the attending physician.

For instance, a young patient may place high value on the removal of a nodule that could potentially be malignant if it increases their chances of a cure and allows them to care for their young children. Conversely, an elderly patient, in whom the nodule was incidentally discovered during an examination for another reason, might prefer to avoid surgery due to the risks associated with the intervention itself, such as anesthesia.

The balance between intervention and non-intervention suggests that both approaches have their own “benefits” and “costs”.

This approach to clinical decision-making, known as the decision curve, diverges from the traditional concept where an “optimal” cutoff point is primarily determined based on the model’s discrimination ability.

In our specific case, it involves determining the optimal point at which the benefits of intervening for a patient with a thyroid nodule outweigh the benefits of not intervening.

Based on clinical, analytical, and ultrasound characteristics, our predictive model estimates the probability or risk that the nodule being evaluated is cancerous.

At one end of the spectrum, we might encounter a patient with a 0.5% risk of malignancy. It seems reasonable to think that, in this case, both the patient and the physician would opt not to intervene. On the other end, we might be evaluating a patient for whom the model predicts a 99% risk of malignancy. In this second case, the physician would advise, and the patient would agree to the excision of the nodule. The same logic would apply if the predicted risks were 2% or 98%. If we continued to narrow the range, we would eventually reach a point where the physician would no longer be certain of their decision.

In our specific case, if the predicted risk were 10%, we would need to perform 10 thyroidectomies to find 1 thyroid cancer. In other words, if the risk is 10%, the odds would be 1:9. If this is the chosen threshold, the physician is implicitly assuming that missing one thyroid cancer is 9 times worse than performing an unnecessary thyroidectomy. In a way, this could be interpreted as the “number-needed-to-intervene,”meaning that a 10% risk would correspond to a number-needed-to-intervene of 10.

In our decision curve (Figure 4), we can see that when our model predicts a risk lower than 9%, the potential benefit does not surpass the strategy of intervening on all patients. Therefore, we should not use our model to make decisions when the predicted risk is lower than 9%.

On the other hand, it is important to understand that the unit of net benefit (the *y*-axis of the decision curve) is true positives. For example, if the curve shows a net benefit of 0.10, this means that at that point we will obtain 10 true positives for every 100 patients. In our specific case, we can see that for a risk threshold of 25%, using our model, the net benefit is 0.10 (10 true positives per 100 patients).

Our study has certain limitations. Alternative models employing methodologies distinct from logistic regression, such as machine learning models, may provide enhanced predictive capacity, which we plan to investigate in future research. Second, the predictive capacity of our model could vary, depending on the sonographer’s experience, considering that there is a high correlation between the experience of the observer and the accuracy of ultrasound evaluation of thyroid nodules [41]. For the last, our predictive model could enhance its accuracy by exclusively analyzing cases of benign thyroid nodules versus classic papillary thyroid cancer nodules, thereby excluding nodules with follicular cancer, whose ultrasound appearance is often indistinguishable from non-cancerous nodules. However, in our population, nodules with follicular cancer exhibit a significantly higher risk of malignancy compared to benign nodules. Therefore, the model could assist in identifying these cases, thereby aiding clinicians in their decision-making process.

We believe that our predictive model can be useful in routine clinical practice as it is based on variables commonly used in daily clinical practice, has undergone an external validation process, and is available online.

## 5. Conclusions

The external validation process of the predictive model for thyroid nodule malignancy risk, developed by our group and available at the link https://obgynreference.shinyapps.io/calccdt/ (accessed on 7 February 2025), demonstrates an adequate capacity to discriminate between malignant and benign nodules. Furthermore, the decision curve analysis conducted indicates that its use can be beneficial in clinical practice.

## Figures and Tables

**Figure 1 diagnostics-15-00686-f001:**
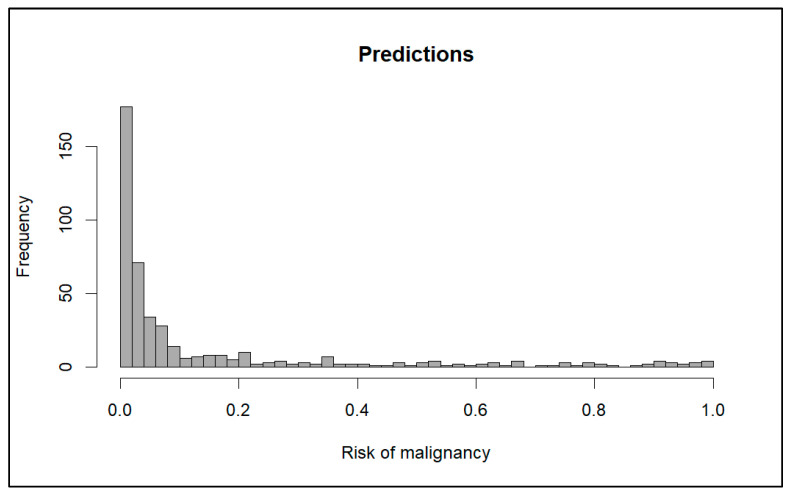
Distribution of probabilities predicted by the model.

**Figure 2 diagnostics-15-00686-f002:**
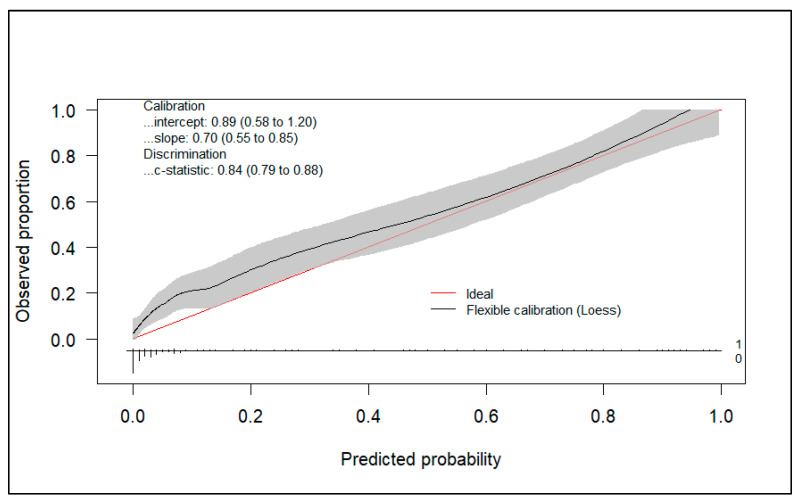
Calibration plot.

**Figure 3 diagnostics-15-00686-f003:**
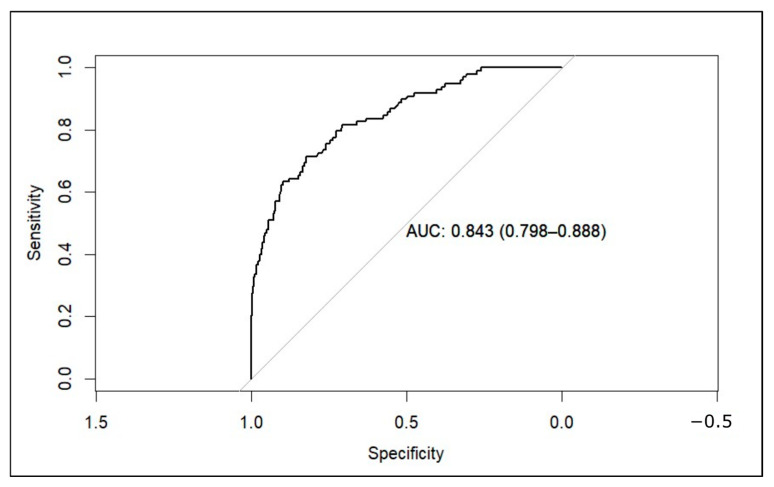
ROC curve of the model applied to the external validation dataset (AUC: area under the curve).

**Figure 4 diagnostics-15-00686-f004:**
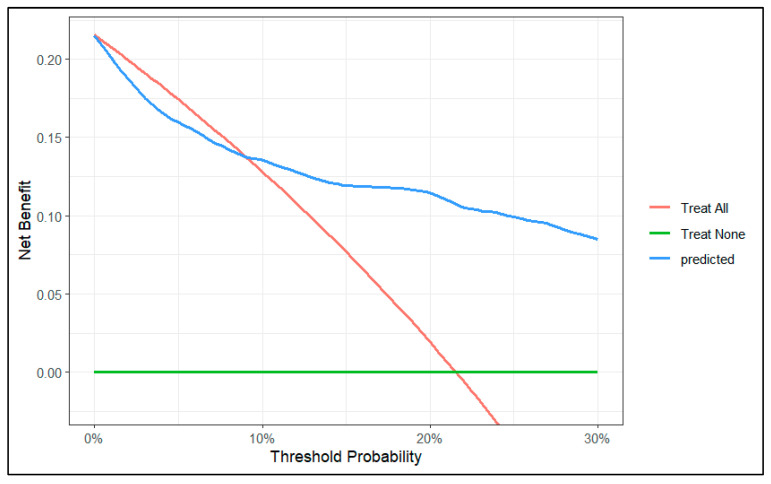
Decision curve of the model between 0 and 30% risk.

**Figure 5 diagnostics-15-00686-f005:**
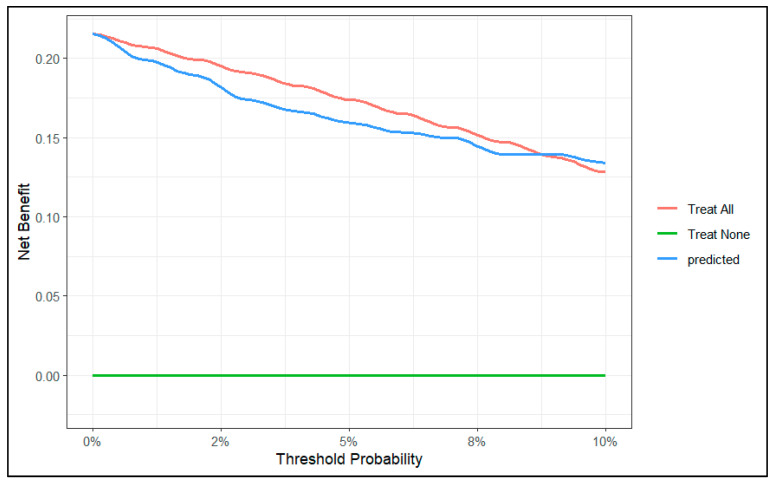
Decision curve of the model between 0 and 10% range of probability.

**Table 1 diagnostics-15-00686-t001:** Summary of the thyroid cancer risk predictive model.

	Estimates	SE	Adjusted OR	95%CI
(lntercept)	−0.09	1.78	0.91	0.02–26.64
Family history of TC	0.84	0.65	2.32	0.65–8.48
Gender (male)	0.66	0.39	1.95	0.89–4.23
Age	−0.18	0.07	0.83	0.72–0.95
Squared age	0.001	0.00	1.001	1.00–1.00
TSH between 0 and 0.369 mcU/mL	−1.45	0.53	0.23	0.08–0.63
TSH higher than 4.701 mcU/mL	0.68	0.61	1.98	0.57–6.44
Autoimmune thyroiditis	0.95	0.35	2.60	1.31–5.25
Solid nodule	1.98	0.77	7.26	1.96–47.64
Suspicious adenopathies	1.05	0.46	2.88	1.19–7.21
Hypoechoic nodule	1.60	0.39	4.96	2.35–11.02
Margins microlobed or irregular	1.25	0.39	3.49	1.64–7.57
Macrocalcifications	0.66	0.56	1.95	0.63–5.68
Microcalcifications	1.40	0.37	4.06	1.98–8.43
Taller than wide nodule	0.66	0.41	1.95	0.86–4.39

SE: standard error; OR: odds ratio; CI: confidence interval; TC: Thyroid cancer.

**Table 2 diagnostics-15-00686-t002:** Clinical, analytical, and sonographic characteristics of studied patients.

Characteristics	Total (*n* = 455)	Benign Nodules (*n* = 357)	Malignant Nodules (*n* = 98)	*p*
Clinical characteristics				
Age (years) (median (IQR))	52 (18)	53 (18)	49 (19.2)	<0.05
Gender *n* (%)				<0.001
Female	366 (80.7%)	300 (84%)	66 (67.3%)	
Male	89 (19.3%)	57 (16%)	32 (32.7%)	
Family history of TC *n* (%)	15 (3.3%)	9 (2.5%)	6 (6.1%)	0.10
Analytical characteristics				
TSH (mcU/mL) (median (IQR))	1 (1.5)	0.9 (1.4)	1.6 (1.59)	<0.001
Autoimmune thyroiditis *n* (%)	93 (20.4%)	60 (16.8%)	33 (33.7%)	<0.001
US characteristics				
Maximum diameter of nodule (mm) (median (IQR))	32 (20)	35 (17)	21 (19.5)	<0.001
Consistency *n* (%)				<0.001
Solid	350 (76.9%)	256 (71.7%)	94 (95.9%)	
Mixed of spongiform	102 (22.4%)	98 (27.5%)	4 (4.1%)	
Cystic	3 (0.7%)	3 (0.8%)	0 (0.0%)	
Echogenicity *n* (%)				<0.001
Hypoechoic	158 (34.7%)	87 (24.4%)	71 (72.4%)	
Iso/Hyperechoic	294 (64.6%)	267 (74.8%)	27 (27.6%)	
Anechoic	3 (0.7%)	3 (0.8%)	0 (0.0%)	
Margins *n* (%)				<0.001
Regular	407 (89.5%)	345 (96.6%)	62 (63.3%)	
Microlobed or irregular	48 (10.5%)	12 (3.4%)	36 (36.7%)	
Shape *n* (%)				<0.05
Wider than tall	425 (93.4%)	339 (95.0%)	86 (87.8%)	
Taller than wide	30 (6.6%)	18 (5%)	12 (12.2%)	
Calcifications *n* (%)				
None	359 (78.9%)	308 (86.3%)	51 (52%)	<0.001
Microcalcifications	51 (11.2%)	14 (3.9%)	37 (37.8%)	
Macrocalcifications	45 (9.9%)	35 (9.8%)	10 (10.2%)	
Suspicious adenopathies *n* (%)	31 (6.8%)	7 (2%)	24 (24.5%)	<0.001

IQR: Interquartile range, TC: Thyroid cancer.

## Data Availability

The raw data supporting the conclusions of this article will be made available by the authors on request.

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
