# Peer review of "External Validation of a Predictive Model for Thyroid Cancer Risk with Decision Curve Analysis"

_diagnostics, 2025, doi:10.3390/diagnostics15060686_

Round 1
Reviewer 1 Report
Comments and Suggestions for Authors
The authors use a large cohort of thyroid lesions to validate their previously developed predictive model that could assess the risk of malignancy in a thyroid nodule. The study is well-planned and presented but there are a few concerns:
1. The authors have included PTC as well as follicular thyroid carcinoma in their cohort and assess their USG findings under the same category 'malignant'. These tumors do not have overlapping USG findings and should be kept separate. Separating them may improve the statistics results further.
2. Low-risk neoplasms like NIFTP must be included in the study as these are the ones where there is a significant overlap with both benign and malignant thyroid entities, posing significant difficulties in diagnosis & management at all levels.
3. Certain subtypes like classic PTC and tall cell PTC will have more diagnostic USG findings. If the cohort is rich in such subtypes, there is a chance of generating a bias. Similarly, a widely invasive follicular cancer (FTC) will have different radiology than a minimally invasive one. Hence, details of subtypes of PTC and FTC should be provided.
Author Response
Dear reviewer,
Thank you for your valuable comments.
Comments 1: The authors have included PTC as well as follicular thyroid carcinoma in their cohort and assess their USG findings under the same category 'malignant'. These tumors do not have overlapping USG findings and should be kept separate. Separating them may improve the statistics results further.
Thank you for your comment.
The primary objective of our model is to assess the risk of malignancy in thyroid nodules to support clinical decision-making. Our findings indicate that, although the malignancy risk of nodules with follicular thyroid carcinoma is lower than that of those with classic papillary carcinoma, it remains significantly higher than that of benign nodules. In our study population, nodules with follicular carcinoma exhibited an average malignancy risk of 26.7 ± 28.2%, compared to 67.5 ± 27.6% for nodules with papillary carcinoma and 9.5 ± 16.2% for benign nodules. Furthermore, 31.6% of nodules with follicular carcinoma were classified as high risk based on ultrasound criteria (American Thyroid Association 2015), in contrast to 82.8% of nodules with classic papillary carcinoma and 8.6% of benign nodules (Carral F, Fernández JJ, Ayala MC et al. Estimation of the risk of malignancy of thyroid nodules by histological subtypes of thyroid cancer. Poster presentation at the 63rd Congress of the Spanish Society of Endocrinology and Nutrition. Las Palmas de Gran Canaria, October 2022).
While it is true that our predictive model would yield improved results by exclusively analyzing benign nodules versus nodules with papillary cancer (thereby excluding cases of follicular cancer), this approach would hinder our ability to assist clinicians in decision-making by omitting a significant number of patients with follicular thyroid cancer. Furthermore, we have demonstrated that our predictive model is also effective for these patients.
Lastly, within the limitations section of our study, we have included the following statement: “…Our predictive model could enhance its accuracy by exclusively analyzing cases of benign thyroid nodules versus classic papillary thyroid cancer nodules, thereby excluding nodules with follicular cancer, whose ultrasound appearance is often indistinguishable from non-cancerous nodules. However, in our population, nodules with follicular cancer exhibit a significantly higher risk of malignancy compared to benign nodules. Therefore, the model could assist in identifying these cases, thereby aiding clinicians in their decision-making process.”
Comments 2: Low-risk neoplasms like NIFTP must be included in the study as these are the ones where there is a significant overlap with both benign and malignant thyroid entities, posing significant difficulties in diagnosis & management at all levels.
NIFTP is a non-invasive follicular thyroid neoplasm with low biological aggressiveness that was previously classified as encapsulated follicular variant papillary thyroid cancer (or well-demarcated) without capsular or vascular invasion, or as follicular tumors of uncertain malignant potential. The NIFTP terminology for low-risk tumors was proposed by the WHO in 2018, although it was not adopted by the Pathology Department of our Center until 2020. Since the development and internal validation of our predictive model was based on information from patients operated on between 2013 and 2018, these patients were classified in the malignant nodules category. In the current sample, only two patients operated on in 2021 and 2022 were classified in the NIFTP category, being classified as benign nodules.
Comments 3: Certain subtypes like classic PTC and tall cell PTC will have more diagnostic USG findings. If the cohort is rich in such subtypes, there is a chance of generating a bias. Similarly, a widely invasive follicular cancer (FTC) will have different radiology than a minimally invasive one. Hence, details of subtypes of PTC and FTC should be provided.
We do not have specific data on the tall cell variant of papillary carcinoma. However, in our population, we have evaluated both the ultrasound characteristics and the estimation of malignancy risk in patients with benign nodules versus different histological subtypes of thyroid cancer. In our population, the high ultrasound risk of malignancy (ATA 2015) was established at 100% for anaplastic or poorly differentiated cancers, compared to 82.8% for classic papillary cancer, 50% for follicular variant papillary cancer, 41% for other variants of papillary cancer, 31.6% for follicular cancer, and 8.6% for benign nodules. The average malignancy risk in anaplastic or poorly differentiated cancer was 69.6 ± 27.0%, compared to 67.5 ± 27.6% for classic papillary cancer, 45.5 ± 36.5% for follicular variant papillary cancer, 45.8 ± 31.2% for other variants of papillary cancer, and 9.5 ± 16.2% for benign nodules, with all differences being statistically significant compared to benign nodules. This supports the use of our predictive model for distinguishing benign thyroid nodules from those with thyroid cancer (Carral F, Fernández JJ, Ayala MC et al. Estimation of the risk of malignancy of thyroid nodules by histological subtypes of thyroid cancer. Poster presentation at the 63rd Congress of the Spanish Society of Endocrinology and Nutrition. Las Palmas de Gran Canaria, October 2022).

Reviewer 2 Report
Comments and Suggestions for Authors
Dear Authors,
Although promising there are some aspects of the manuscript that require your attention.
Table 1, the family history of TC has a positive risk for malignancy, but subsequently, you say that the p-value is not significantly statistical. I believe that this is due to the very few number of cases of familial carcinoma in your study group. Please expand on this subject.
Table 2, if there are only 15 cases with a family history of TC, why do you choose to keep them?
Figure 1, please include the actual values on each column.
Figure 3 captions, is the ROC curve of what?
In the Discussion section, you need to underline the need for early detection of TC compared with the limited prognosis of advanced thyroid tumors. One reference could be https://doi.org/10.3390/biomedicines12102204.
Please include all the abbreviations in the list at the end of the manuscript.
Please format the references according to MDPI instructions for authors.
Looking forward to receiving the improved version of your manuscript.
Author Response
Dear Reviewer,
Thank you very much for your valuable observations. Below, we present our responses to your comments.
Comments 1. Table 1, the family history of TC has a positive risk for malignancy, but subsequently, you say that the p-value is not significantly statistical. I believe that this is due to the very few number of cases of familial carcinoma in your study group. Please expand on this subject.
Thank you for your comment. The adjusted odds ratio (OR) obtained from the logistic regression model was indeed 2.32. However, the 95% confidence interval ranged from 0.65 to 8.68. As you correctly noted, this lack of statistical significance may be attributed to the small number of cases with a family history of thyroid cancer in our previous study. Nevertheless, scientific evidence indicates that a family history of thyroid cancer is a risk factor for the disease. Therefore, we chose to retain this variable in our model, observing a slight increase in predictive capacity as a result.
Comments 2. Table 2, if there are only 15 cases with a family history of TC, why do you choose to keep them?
Thank you again for the comment. We decided to retain these cases because, in the original predictive model, the variable 'family history of TC' is one of the predictor variables. We believe that this approach enhances the robustness of the external validation, as it is conducted with real-world data.
Comments 3. Figure 1, please include the actual values on each column.
Thank you for your comment. Due to an error, this histogram was initially constructed showing percentages instead of frequencies, despite manually specifying 'Frequency' on the y-axis label. This mistake has been corrected in the figure, and the y-axis scale are now accurate. For the sake of clarity, we have decided not to specify the value of each of the 50 columns that make up the histogram.
Comments 4. Figure 3 captions, is the ROC curve of what?
Thank you for your comment. We have added 'ROC curve of the model applied to the external validation dataset' to Figure 3 captions.
Comments 5. In the Discussion section, you need to underline the need for early detection of TC compared with the limited prognosis of advanced thyroid tumors. One reference could be https://doi.org/10.3390/biomedicines12102204.
Thanks you for your suggestions. We have added the next paragraph to the Discussion section:
The early detection of thyroid cancer (TC) is crucial for improving patient outcomes, particularly when compared to the limited prognosis associated with advanced thyroid tumors. Early diagnosis allows for timely intervention, which significantly enhances survival rates, especially in cases like medullary thyroid carcinoma (MTC), where early-stage detection can lead to a 90-100% ten-year survival rate. In contrast, advanced stages of TC are linked to a stark decline in prognosis, with survival rates dropping to as low as 17% (Lubin D, Sadow PM. Development and validation of an RNA sequencing–based classifier for medullary thyroid carcinoma on thyroid FNA. Cancer Cytopathology. 2022;131(3):154–157, Vrinceanu, D.; Dumitru, M.; Marinescu, A.; Serboiu, C.; Musat, G.; Radulescu, M.; Popa-Cherecheanu, M.; Ciornei, C.; Manole, F. Management of Giant Thyroid Tumors in Patients with Multiple Comorbidities in a Tertiary Head and Neck Surgery Center. Biomedicines 2024, 12, 2204. https://doi.org/10.3390/biomedicines12102204)
Comments 6. Please include all the abbreviations in the list at the end of the manuscript.
Thank you for the suggestion. We have reviewed the entire manuscript and added the missing abbreviations at the end.
Comments 7. Please format the references according to MDPI instructions for authors.
Thank you for the warning. We have reviewed all the references and believe they are now correctly cited in the format required by MDPI.